# Information differences across spatial resolutions and scales for disease surveillance and analysis: The case of Visceral Leishmaniasis in Brazil

**Joseph L. Servadio**[1]*, **Gustavo Machado**[2], **Julio Alvarez**[3,4], **Francisco Edilson de Ferreira Lima Júnior**[5], **Renato Vieira Alves**[5], **Matteo Convertino**[6]

1 Division of Environmental Health Sciences, University of Minnesota School of Public Health, Minneapolis, Minnesota, United States of America, 2 Department of Population Health and Pathobiology, College of Veterinary Medicine, North Carolina State University, Raleigh, North Carolina, United States of America, 3 VISAVET Health Surveillance Center, Universidad Complutense, Madrid, Spain, 4 Departamento de Sanidad Animal, Facultad de Veterinaria, Universidad Complutense, Madrid, Spain, 5 Secretaria de Vigilância em Saúde, Ministério da Saúde (SVS-MH), Brasília, Brazil, 6 Nexus Group, Graduate School of Information Science and Technology and GI-CoRE Station for Big-Data and Cybersecurity, Hokkaido University, Sapporo, Hokkaido, Japan

* serva024@umn.edu

**Data Availability Statement:** All relevant data are within the manuscript and its Supporting Information files. They can be found in S3_File.zip.

## Abstract

Nationwide disease surveillance at a high spatial resolution is desired for many infectious diseases, including Visceral Leishmaniasis. Statistical and mathematical models using data collected from surveillance activities often use a spatial resolution and scale either constrained by data availability or chosen arbitrarily. Sensitivity of model results to the choice of spatial resolution and scale is not, however, frequently evaluated. This study aims to determine if the choice of spatial resolution and scale are likely to impact statistical and mathematical analyses. Visceral Leishmaniasis in Brazil is used as a case study. Probabilistic characteristics of disease incidence, representing a likely outcome in a model, are compared across spatial resolutions and scales. Best fitting distributions were fit to annual incidence from 2004 to 2014 by municipality and by state. Best fits were defined as the distribution family and parameterization minimizing the sum of absolute error, evaluated through a simulated annealing algorithm. Gamma and Poisson distributions provided best fits for incidence, both among individual states and nationwide. Comparisons of distributions using Kullback-Leibler divergence shows that incidence by state and by municipality do not follow distributions that provide equivalent information. Few states with Gamma distributed incidence follow a distribution closely resembling that for national incidence. These results demonstrate empirically how choice of spatial resolution and scale can impact mathematical and statistical models.

## 1. Introduction

Infectious disease research often relies on data generated through passive or active surveillance activities, which can suffer from important limitations due to variation in methods and

**Funding:** This work was funded by the Academic Health Center Faculty Research Development Grant Program (FRD no. 16.36) from the University of Minnesota Twin-Cities, received by JA and MC. The funders played no role in study design, data collection and analysis, decision to publish, or preparation of the manuscript.

**Competing interests:** The authors have declared that no competing interests exist.

capacities for data collection [1, 2]. Typically, researchers aim to collect data at a high spatial resolution, that is, in the form of small surveillance units such as counties or municipalities rather than states or nations [3, 4], though this may not always be seen as beneficial [5]. Conducting surveillance at a high spatial resolution, however, is often unrealistic when considering large areas and constrained resources [6, 7].

Data collected from infectious disease surveillance activities are often used in research involving mathematical or statistical models. In such analyses, matters related to data quality are of concern. The choice of spatial resolution is often based on data availability or chosen arbitrarily, with little attention given to whether this decision may impact model results. Aggregating data into larger spatial units can aid in computational efficiency, but creates the risk of introducing ecological fallacy [8] and masking heterogeneity within those larger units if conclusions are drawn inappropriately [9–11]. This would be particularly problematic when aiming to seek disease etiologies. This concept is related to the modifiable areal unit problem [12, 13] in its discussion of choices of spatial units impacting results. The modifiable areal unit problem is always present when using spatial data, but is infrequently acknowledged and rarely quantified [13]. The choice of spatial resolution may impact any models used; previous studies using mathematical or statistical models have investigated the importance of high resolution data by repeating analyses using data at different resolutions and then comparing results [10, 11, 14].

An additional challenge to high quality surveillance is the need for surveillance over a large spatial scale, referring to the entire area where surveillance is being conducted. Here, spatial scale differs from spatial resolution in their definitions as follows: spatial scale refers to the total spatial area being examined, while spatial resolution refers to the size of the individual spatial units within that area. Large-scale surveillance can be particularly challenging for nations with large land areas and populations. In these circumstances, there is potential benefit in identifying a smaller area, such as a state or group of states, where surveillance can adequately estimate the national disease burden. The characteristic of having smaller areas representative of the whole for a large range of sizes is known as scale invariance or fractality [15]. Scale invariance is ubiquitous in many socio-ecological patterns such as finance [16], ecology [17], biochemical processes [18], and biology across time scales [19].

Scale invariance is an infrequently examined concept in infectious disease surveillance and epidemiology in general, though it has relevance to many forms of data analysis or modeling. In research involving statistical or mathematical models, the scale used, whether an entire nation, portion of a nation, or other extent, may impact the structure and products of the model. Scale invariance in infectious disease research is more frequently used to describe scale-free networks, typically applied to human communicable diseases [20] or transmission paths of infectious diseases [21]. For practical purposes in epidemiology, identifying smaller regions that represent a larger area or even an entire nation could allow the design of targeted surveillance strategies and conserve resources [22]. Even in the absence of true scale invariance, self-similarity can be observed [15] where some smaller areas can be used to describe the whole. In other situations, however, the spatial scale of interest impacted the physical processes being studied [23].

The topics of spatial resolution and scale are relevant for research pertaining to numerous health outcomes. Here, Visceral Leishmaniasis is examined as a case study. Visceral Leishmaniasis (VL), caused by a *Leishmania infantum* parasite (known in Latin America as *Leishmania chagasi*) [24], is the most severe form of Leishmaniasis and is fatal in the vast majority of untreated cases [25]. The parasite is typically transmitted from an infected to a non-infected host through the bites of phlebotomine sand flies [26]. Visceral Leishmaniasis presentation can include symptoms such as fever, enlargement of the spleen and liver, and anaemia [25]. It

is estimated that between 200,000 and 400,000 cases of VL occur worldwide annually, with approximately ten percent being fatal [27].

Brazil is one of ten world nations with the greatest VL burdens [25], with the remaining nations located primarily in East Africa and South and Southeast Asia [25]. Estimates report that 90% of VL cases that occur in the Americas occur in Brazil [25] where, canines are a disease reservoir [28–30]. The estimated age adjusted incidence rate from VL in Brazil is 1.84 cases per 100,000 population, and the mortality rate from VL in Brazil is 0.15 deaths per 100,000 population, with approximately eight percent of cases being fatal [31]. Areas of Brazil that previously had accounted for only 15% of all cases reported nationally now can see nearly half of the nation's cases [30]. The disease has also become more common in urban areas in recent decades [28, 32, 33], making it a major public health concern and an important target for surveillance programs. As of 2015, based on the data used for this study, the Federal District and 18 of the 26 states in Brazil meet the criterion of being an endemic state for VL, which is seeing at least one case in all three previous years [34]. The states that were not endemic at the time are Acre, Amapá, Amazonas, Espírito Santo, Paraná, Rondônia, Roraima, and Santa Catarina [34, 35]. As of 2019, Espírito Santo and Paraná became endemic states [35].

This study aims to assess the potential impact of using different spatial resolutions and scales on statistical and mathematical models using surveillance data applied to VL cases in Brazil. In order to do so, two objectives are pursued: (1) to determine if surveillance using incidence by state or municipality leads to different distributional fits of disease incidence; and (2) to determine if conducting VL surveillance on a region within Brazil would equivalently characterize the nation's VL incidence. This is done by using best fitting probability distributions to describe disease data without incorporating outside information. A conceptual visualization of the study aims is presented in Fig 1. Prior to conducting statistical analyses or models, researchers may need to decide whether to consider data using different spatial resolutions as well as the scale of analysis; the results of this study will provide insight into whether the subsequent results may be sensitive to this decision.

## 2. Methods and materials

### 2.1 Study setting and data

The setting for this study is Brazil, the largest nation in South America in both land area and population. Case data were provided by the Brazilian Ministries of Health [36] and include VL case counts by municipality nationwide, with the exception of the Federal District, totaling 26 states and 5,561 municipalities. The Federal District was excluded because it is not a state with multiple municipalities, and therefore cannot be aggregated to differentiate between the municipality resolution and the state resolution. Yearly case counts by municipality between 2004 and 2014 are reported. Annual populations for each municipality are publicly available through the Brazilian Institute of Geography and Statistics [37] to calculate annual incidence, discretized to represent cases per 100,000 population per year. Population data were available for all years except 2007 and 2010. In these two years, the arithmetic means of the populations of the two adjacent years were used in place of the missing populations. Population data were available for 5,538 of the municipalities with VL data, providing the final sample for this study.

### 2.2 Inferring probability distributions

This study compares spatial resolutions and scales using probability distributions rather than by fitting models with assumptions and conducting a sensitivity analysis. This was done to avoid imposing assumptions of a particular model, keeping the examination of scale and resolution as the focus of analyses with regards to the characteristics of VL incidence itself rather

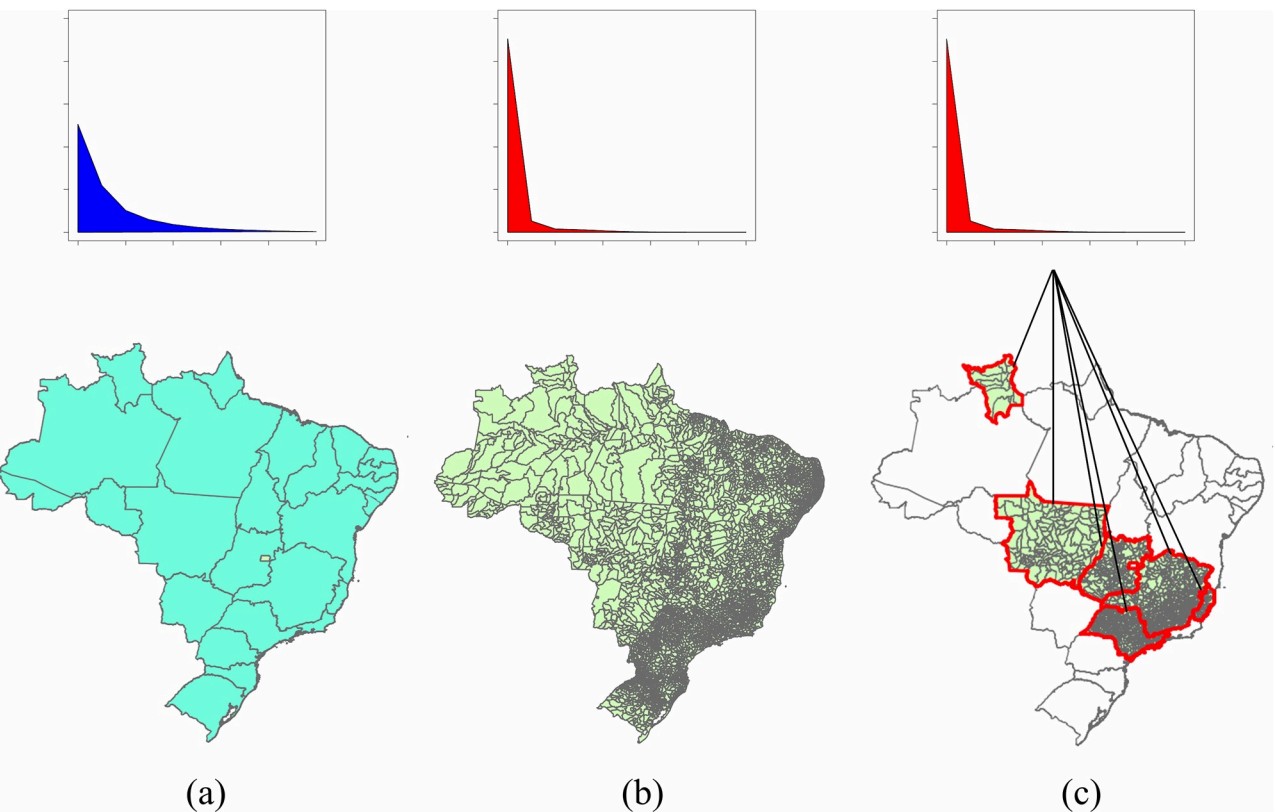

**Fig 1. Graphical overview of the study objectives: (a) fit distribution to annual incidence of Visceral Leishmaniasis (VL) by state, (b) fit distribution to annual incidence of VL by municipality, (c) fit distributions to annual incidence of VL by municipality within each state.** Comparisons of these fitted distributions indicate whether characterizing VL incidence by state or municipality are equivalent, impacting statistical analyses using these data.

than the relationship between incidence and other data. Best fitting distributions from multiple considered distributional families were fit for (1) annual incidence for each individual state using the municipality as the unit of surveillance; (2) annual incidence nationwide using the municipality as the unit of surveillance; and (3) annual incidence nationwide using the state as the unit of surveillance (Fig 1). All 11 years of observation were included.

Common candidate distributions were selected based on exploratory analyses, including visual analyses, quantiles, and summary statistics, and having a nonnegative support; wide ranges of parameters for each distribution were tested. The Poisson, Zero Inflated Poisson (ZIP), and Zero One Inflated Poisson (ZOIP) [38] distributions were selected as candidate distributions along with the Gamma, Exponential, Power Law, and Uniform distributions rounded to fit discrete data. These are described in Table 1.

Each distribution was evaluated for the optimal parameter set that minimizes the sum of absolute error (SAE), defined as

$$SAE = \Sigma_x |p(x) - p_{obs}(x)| \tag{1}$$

where $p(x)$ represents the probability of observing an incidence of $x$ cases per 100,000 person-years based on the proposed distribution indicated in Table 1 and $p_{obs}(x)$ represents the

**Table 1. Candidate distributions used for fitting distributions.**

| Distribution | Probability mass function |
|---|---|
| Poisson | $P(X = x) = \frac{\theta^x}{x!} \exp(-\theta)$ |
| Zero Inflated Poisson (ZIP) | $P(X = 0) = \alpha + (1 - \alpha)\exp(-\theta)$ <br> $P(X = x) = (1 - \alpha) \frac{\theta^x}{x!} \exp(-\theta),\ x > 1$ |
| Zero One Inflated Poisson (ZOIP) | $P(X = 0) = \alpha + (1 - \alpha - \beta)\exp(-\theta)$ <br> $P(X = 1) = \beta + (1 - \alpha - \beta)\,\theta\exp(-\theta)$ <br> $P(X = x) = (1 - \alpha - \beta) \frac{\theta^x}{x!} \exp(-\theta),\ x > 1$ |
| Rounded Exponential | $P(X = x) = \int_0^{x+0.5} f(y)\,dy - \int_0^{x-0.5} f(y)\,dy$ <br> $f(y) = \theta \exp(-\theta\,y)$ |
| Rounded Gamma | $P(X = x) = \int_0^{x+0.5} f(y)\,dy - \int_0^{x-0.5} f(y)\,dy$ <br> $f(y) = \frac{\beta^\alpha}{\Gamma(\alpha)}\, y^{\alpha-1} \exp(-\beta\,y)$ |
| Rounded Power Law | $P(X = x) = \int_0^{x+0.5} f(y)\,dy - \int_0^{x-0.5} f(y)\,dy$ <br> $f(y) = \frac{\alpha-1}{x_{min}} \left(\frac{x}{x_{min}}\right)^{-\alpha},\ x > x_{min}$ |
| Rounded Uniform | $P(X = x) = \frac{1}{b-a},\ x \in \mathbb{Z} \cap [a,\ b]$ |

observed proportion of incidence values equaling $x$. This measure compares similarities between the proposed distributions and observed data and is less sensitive to outliers than other measures [39].

The optimal parameter set for each distribution was found through simulated annealing [40], an optimization algorithm based on Monte Carlo sampling. The algorithm was run with three chains for 50,000 simulations to assure convergence. Convergence was reached if each of the three chains produced identical parameter values, indicating a lack of movement to other parameter values, for at least 500 iterations, as well as one of the following hierarchical criteria: (1) final parameter values across pairs of chains had an absolute difference of less than 0.01; (2) final parameter values had an absolute difference of less than 0.1 and associated SAE values had an absolute difference of less than 0.01; (3) SAE values had an absolute difference of less than 0.001. The second and third criteria were necessary due to some parameterizations having very similar SAE values. If convergence was not reached in 50,000 iterations, the three chains were restarted with the parameterization that led to the lowest SAE value in the chain as initial values, and the simulated annealing algorithm was repeated, increasing the simulation count by 5,000. This was repeated until convergence was reached.

## 2.3 Comparing distributions

**2.3.1 Sensitivity to spatial resolution.** The first aim of comparing distributions is to determine if changing the spatial resolution alters the distributional fit of incidence. In future modeling studies, differences in distributional fit could lead to changes in model outputs as a result of the spatial resolution of the data, whether aggregated by choice or through surveillance availability. This was done by comparing the fitted state-resolution distribution to an expected state-resolution distribution for the nation based on the fitted municipality-resolution distribution for the nation. This expected distribution was generated empirically by drawing Monte Carlo samples from the fitted municipality-resolution distribution.

If incidence is denoted by $X$ as a random variable following the fitted municipality-resolution distribution for the nation; states are denoted by $s$; state $s$ has $n_s$ municipalities, denoted by $m$; and municipality $m$ has a population of $p_{my}$ in year $y$, this empirical distribution was

generated by drawing 1,000 samples of

$$z = \frac{\sum_{m=1}^{s} p_{my} x_m}{\sum_{m=1}^{s} p_{my}} \tag{2}$$

for each state under each year of observation. The numerator of Eq (2) draws values of $X_m$ for each municipality in a state and multiplies the value by the population of the municipality to sample a case count for the municipality. The sum of these is divided by the total state population to produce an expected state-resolution incidence for a year based on the municipality-resolution distribution for incidence. The 1,000 samples of $z$ then produce an empirical distribution.

Relative proportions of incidence values in these simulated values were compared to the probabilities of each incidence value from the fitted state-resolution distribution through Kullback-Leibler (KL) divergence [41]. KL divergence represents the additional information needed when using one distribution to describe data from another distribution. By measuring dissimilarity, it has an inverse relationship to Mutual Information, which represents similarity of variables [41]. Thus, KL divergence is a measure of the Value of Information [42]. For two random variables, denoted A and B, the KL divergence from A to B, compared to the Shannon entropy in the distribution of A, shows a relative increase in information, using bits as units, needed to describe the distribution of B with that of A [41]. This is shown in the ratio of KL divergence to Shannon entropy (denoted $H$), which can be defined as the Required Relative Information Gain (RRIG), where

$$RRIG = \frac{KL_{A \to B}}{H_A} \tag{3}$$

shows the needed increase in information for the distribution of B to describe data from the distribution of A [41]. A value of 1 represents an information increase by 100%, or a doubling of information, though this is not an upper bound. A large RRIG value is indicative of distinct differences between distributions, indicating that characterizations of VL incidence are sensitive to the resolution of surveillance. An RRIG above 5% was selected a priori as a threshold for having a distinct difference in distribution.

**2.3.2 Sensitivity to spatial scale.**   The aim of comparing spatial scales involves comparing the municipality-resolution distributions of each state and of the nation. In the presence of scale invariance, individual states would have the same or similar distributions, which would be similar to the distribution for the nation. Distributions for municipality-resolution case counts for the nation and each state were compared through RRIG.

All analyses were performed using R version 3.6.0 [43]. The 'poweRlaw' package was used to calculate probability density and mass for the Power Law distribution [44].

## 3. Results

Of the 26 states in Brazil, 22 were included in analyses since they all observed more than five nonzero unique annual municipality-resolution incidence values over the study period (S1 Table). The remaining four states were excluded because their incidences over the 11 years did not provide enough unique values to reliably fit a distribution. All 18 endemic states from 2015 [35] were included as well as Espírito Santo, Paraná, Rondônia, and Roraima. Fig 2(a) and 2(b) shows total case counts by state and by municipality over the entire time period.

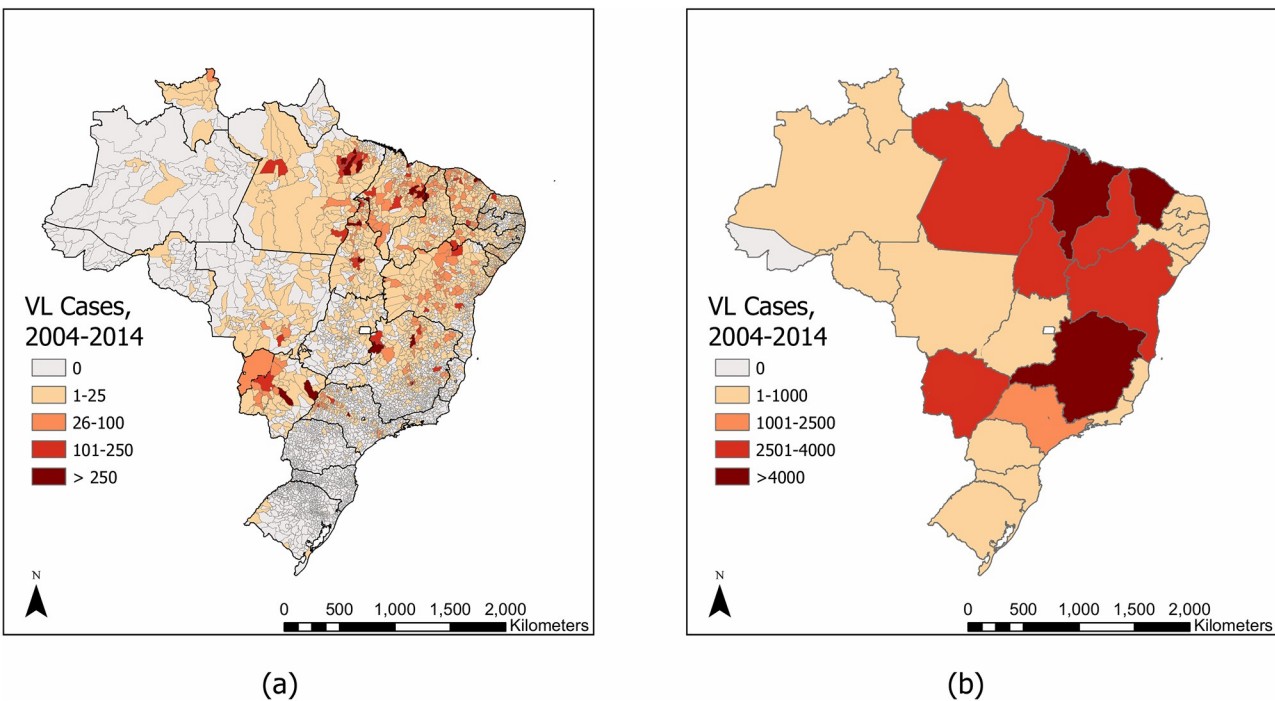

**Fig 2. Total case counts by (a) municipality and (b) state between 2004 and 2014.**

## 3.1 Fitted distributions

The uniform distribution for nationwide municipality-resolution incidence was not able to converge after increasing the iteration count to 200,000. All other distributions converged to optimal values. The best fitting municipality-resolution distributions for individual states varied. Annual incidence values from ten states were best fit by the rounded Gamma distribution, incidences from seven states were best fit by the Poisson distribution, incidences from three states were best fit by the Zero Inflated Poisson distribution, and incidences from two states were best fit by the Zero One Inflated Poisson distribution. Specific parameters are shown by state in Table 2. Plots of the probability mass functions of each state's municipality-resolution distribution are shown in Fig 3. Nationwide, the best fitting distribution for municipality-resolution incidence was the Gamma distribution, and the best fitting distribution for state-resolution incidence was the Zero One Inflated Poisson distribution (Table 2). No notable differences were seen in distributional fit among VL endemic and non-endemic states.

## 3.2 Comparisons across resolutions and scales

The RRIG from Eq (3) was used to quantify the similarities between distributions. The distributions fitted to state-resolution incidence and to the empirical distribution created from the national municipality-resolution distribution and Eq (2) were first compared to determine the sensitivity to the resolution of surveillance data. The RRIG was 0.425, representing a needed increase of information by 42.5% (Table 2). These results are indicative of strong sensitivity to the resolution of surveillance; the distribution for VL incidence by state does not accurately describe the distribution of incidence by municipality.

Comparisons between individual states' municipality-resolution incidence and national municipality-resolution incidence are shown in Table 2 using RRIG from Eq (3). The

**Table 2. Distributions for municipality-resolution incidences by state, municipality-resolution incidence nationwide, and state-resolution incidence nationwide.**
KL/H calculated from Eq (3) shows comparisons of fitted distributions to that for the nationwide municipality-resolution distribution (reference).

| State | Distribution | Mean | Variance | KL/H |
|---|---|---|---|---|
| AL* | Poisson(0.234) | 0.234 | 0.234 | 0.456 |
| BA* | Gamma(0.078, 0.022) | 3.545 | 161.157 | 0.191 |
| CE* | Gamma(0.191, 0.038) | 5.026 | 132.272 | 0.824 |
| ES | Gamma(0.003, 0.002) | 1.500 | 750.000 | 0.084 |
| GO* | Gamma(0.016, 0.016) | 1 | 62.5 | 0.009 |
| MA* | ZOIP(0.599, 0.004, 7.292) | 2.899 | 15.605 | 1.154 |
| MG* | Gamma(0.024, 0.020) | 1.2 | 60 | 0.001 |
| MS* | Poisson(0.378) | 0.378 | 0.378 | 0.791 |
| MT* | Gamma(0.015, 0.003) | 5 | 1666.667 | 0.007 |
| PA* | Gamma(0.094, 0.022) | 4.273 | 194.215 | 0.289 |
| PB* | Poisson(0.077) | 0.077 | 0.077 | 0.155 |
| PE* | ZIP(0.819, 4.193) | 0.759 | 3.365 | 0.263 |
| PI* | ZIP(0.723, 14.44) | 4.000 | 45.759 | 0.878 |
| PR | Poisson(0.003) | 0.003 | 0.003 | 0.151 |
| RJ* | Gamma(0.002, 0.009) | 0.222 | 24.691 | 0.108 |
| RN* | Poisson(0.085) | 0.085 | 0.085 | 0.165 |
| RO | Poisson(0.013) | 0.013 | 0.013 | 0.128 |
| RR | Gamma(0.015, 0) | 15 | 15000 | 0.021 |
| RS* | ZIP(0.998, 4.338) | 0.009 | 0.046 | 0.146 |
| SE* | ZOIP(0.776, 0.006, 5.254) | 1.151 | 5.43 | 0.388 |
| SP* | Gamma(0.014, 0.013) | 1.077 | 82.840 | 0.01 |
| TO* | Poisson(0.450) | 0.450 | 0.450 | 0.960 |
| Brazil (Municipality) | Gamma(0.024, 0.017) | 1.411 | 83.045 | Reference |
| Brazil (State) | ZOIP(0.379, 0.074, 2.82) | 1.617 | 3.353 | 0.425 |

* denotes states considered endemic for Visceral Leishmaniasis in 2015 [34, 35].

nationwide, municipality-resolution distribution is used as the reference for comparisons. The results show that six of the 22 states had incidence following a distribution close to that of the nation (RRIG<0.05) (Table 2, Fig 4). Any of these states could individually describe municipality-resolution incidence of the nation using their own incidence data. Because not all states adequately characterize national burden, true scale invariance was not seen, though self-similarity was seen in the selected states. The states that exhibited some self-similar behaviors all followed a Gamma distribution with generally similar parameters, particularly low values of shape parameters. Many of these states were located near the center of the nation (Fig 4).

## 4. Discussion

This study aimed to assess the importance of the spatial scale and resolution used for VL surveillance and subsequent quantitative analyses. This is also reflective of the dynamics of VL at different scales determined by the distributions of incidence. Probability distributions were fit to incidences at different spatial resolutions and scales and then compared to determine if distributional fit was sensitive to the choice of scale and resolution. Aggregating municipality-resolution incidences into state-resolution incidences led to notably different probabilistic characteristics of disease burden, suggesting the existence of different processes driving disease occurrence at the two resolutions. When continuing surveillance at the municipality resolution, six states' incidences follow distributions that adequately describe those of each other as

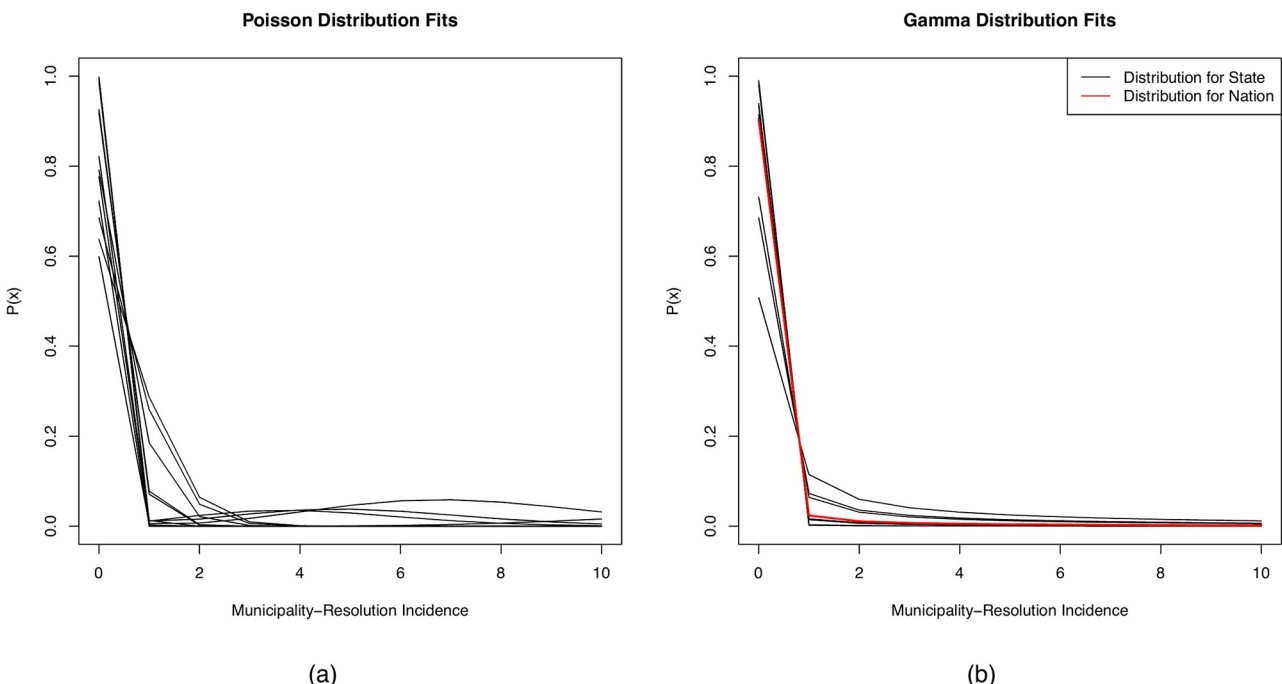

**Fig 3. Probability mass functions for distributions fit to municipality-resolution incidences among states and for the nation.** Incidences were fit by either (a) a Poisson distribution, including distributions with zero and one inflations, or (b) a Gamma distribution.

well as the nation of Brazil. While our results provide evidence against true invariance to resolution and scale, some self-similarity is seen in both distributional parameters and moments. This happens for states that are following a Gamma distribution, which implies medium-long range dispersal of cases and a potential tendency toward a power-law distribution for small scale and shape parameters.

The self-similarity seen in six states does not indicate that significant resources can be saved in Brazil by concentrating surveillance in a smaller area because they are not representative of the other states. The remaining states still need to undergo surveillance in order for their VL burden to be adequately characterized. Furthermore, it is of interest for public health to know where all VL cases occur in order to intervene in an outbreak. If greater self-similarity were seen, it would largely be of interest to researchers who could potentially generalize results of a smaller area to the nation of Brazil through conducting more intensive data collection for additional data in a smaller area. However, because scale invariance was not seen and self-similarity was seen in a small number of states, it is unlikely that descriptions of VL burden in a smaller region of Brazil are generalizable to the entire nation. These considerations consider the current observed state, for instance in case of a widespread propagation of the disease in long range.

Differences in municipality-resolution distributions among states suggest that different factors may influence VL risk across states. Environmental factors shown to influence VL case risk include vector populations, canine cases, precipitation [45], proximity to wooded areas [46], land use, deforestation [47], temperature, and humidity [48]. The question of resolution dependence is targeting whether the elementary unit at which we look into VL dynamics makes a difference for reproducing the distributional representation of VL incidence at the scale of analysis. These natural phenomena related to VL burden may differ across spatial

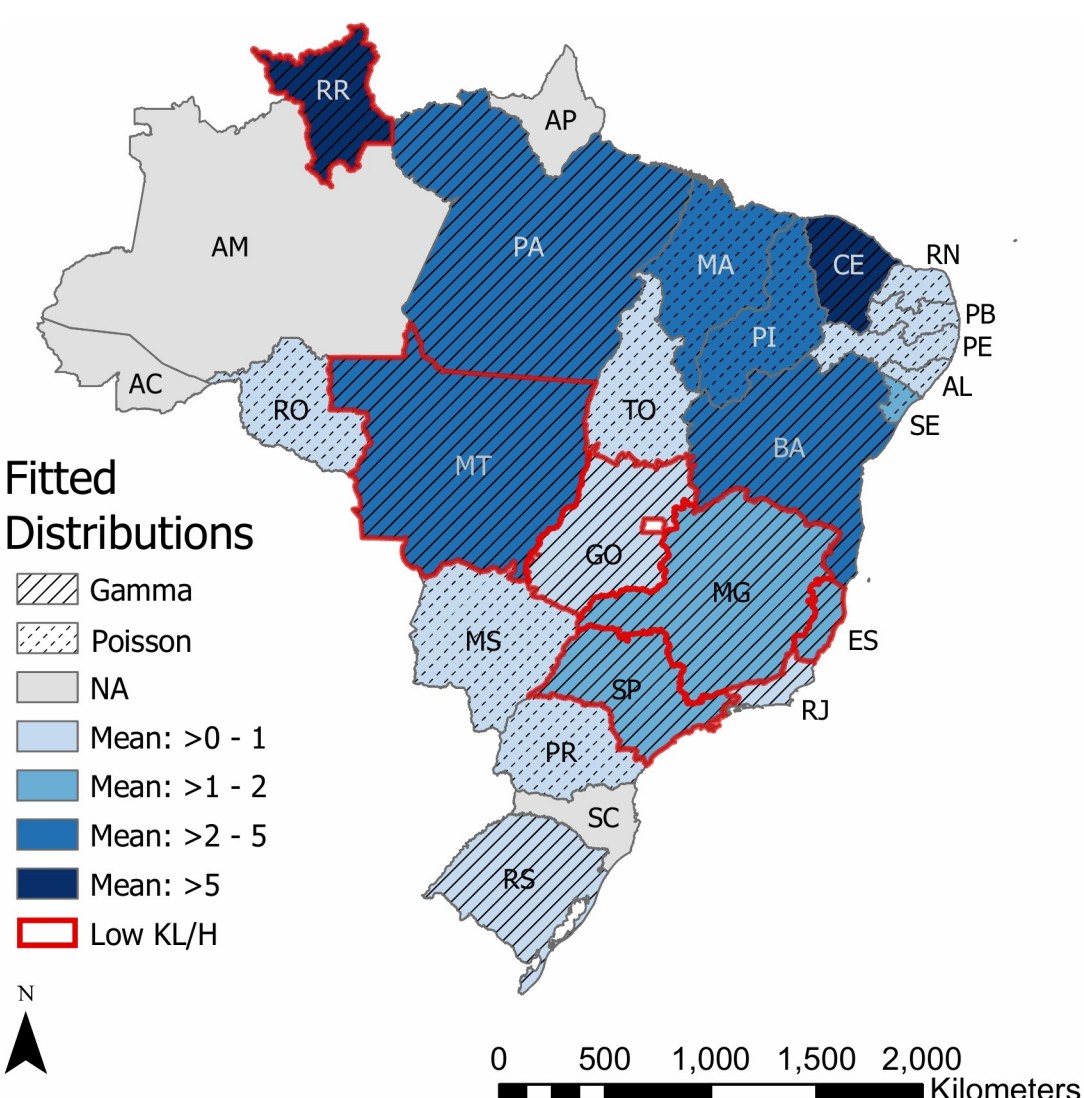

**Fig 4. Expected values and families of fitted municipality-resolution distributions.** States outlined in red had low KL divergence to Shannon entropy ratio with the national municipality-resolution distribution, indicating self-similarity.

scales and resolutions, similarly to other physical phenomena [49]. Though vector populations and precipitation do not explicitly conform to local political boundaries, different regions of Brazil likely see differences in these risk factors through differences in managing socio-ecological factors. These facts and the finding that distributions across states differ (also varying when resolution varies) are important considerations when analyzing disease data. It would be advisable to analyze data for individual locations [45, 48, 50] or use random effects [36].

The results from this study do not necessarily suggest that one spatial resolution is more "correct" than another or favor a particular resolution for analysis. The resolution for future statistical analyses should rely on the research question being posed and desired interpretation of results. However, the resolution dependence implies that, assuming accuracy and precision in assigning municipalities to observed cases, aggregating incidence to the state resolution likely introduces ecological fallacy. Thus, high resolution is likely beneficial to capture disease dynamics accurately. These results also illustrate the intuitive modifiable areal unit problem

quantitatively by providing, through the distributional fits, a way to quantitatively observe the severity of the problem in the application of VL incidence. For high-resolution incidence, the most likely VL dynamics are represented by the Gamma distribution. These considerations should be always taken into account when collecting and analyzing data because they indicate that the choice of resolution will impact model results and their interpretation. Data characterizations and analyses at one resolution are not interchangeable for characterizations and analyses at the other resolution. A related point to note is that diligent surveillance is important when conducted at a finer spatial resolution to ensure accuracy of municipalities that are matched to cases.

This study is, to the authors' knowledge, the first to examine VL incidence for sensitivity to scale and resolution of surveillance data by finding best-fitting distributions to characterize incidence. Other studies have analyzed the fractality of other diseases, such as cholera, and how that is important for a simple estimation of disease spread in term of geography and magnitude [51]. Similar distribution fitting processes are used in veterinary epidemiology [52], but less frequently in human disease epidemiology. This analysis is important for informing future disease burden by providing location-specific estimates of expected annual incidence.

The findings of this study can benefit surveillance, healthcare infrastructure, digital epidemiology, and public health research focused on disease ecology. Care for an individual VL patient in Brazil, including diagnosis, treatment, and medical care, is estimated to be approximately $500 (US) (plus an additional $1470 (US) for secondary prophylaxis among VL patients with HIV) and lasts between seven and 20 days [53]. This is a high individual healthcare cost: yet, designing optimal surveillance that allows public health practitioners to understand and prevent VL is an incredibly valuable task socially and economically. These results and methods (applicable to any disease) can optimize disease data analysis and surveillance for the reduction of the systemic disease burden.

Using only VL incidence data and not introducing other data sources provides focus on what would be the outcome variable of a typical statistical analysis independently of any other predictors that may be introduced. Refitting models at multiple resolutions or scales assumes that the outcome, in this instance VL incidence, follows the same distribution in each scale and/or resolution. For example, using a lognormal regression model with two resolutions assumes that incidence at both resolutions follows a lognormal distribution, which may not be correct. When analyzing municipality-resolution cases, not all states have distributions in the same family, and distributions following the same family have different parameterizations because of the likely differential importance of the underlying socio-environmental drivers. The latter point further motivates the use of Bayesian hierarchical models or other models, for instance statistical physics and/or information theoretic models, which are able to handle the information of scale and resolution controlling factors.

We show that the information theoretic RRIG can determine the amount of information needed to describe the data using different resolutions or scales. It can be used as an information theoretic tool for scaling (downscaling or upscaling, depending on the purpose) epidemiological data considering their value and underlying distributions.

An additional point of novelty is the use of the ZOIP and Gamma distributions to characterize VL incidence. Both distributions are uncommonly used for infectious disease incidence, despite closely fitting observed data. The ZOIP distribution offers the advantage of specifically fitting high frequencies of counts of one, describing single spurious cases. The Gamma distribution is advantageous for placing high probability on low values. More specifically into the statistical physics of disease ecology, the Gamma distribution has similarities to heavy tail distributions (for small shape and scale parameters) and ZOIP represents Poisson distributions highlighting local/random and medium-range disease dynamics. The higher statistical

complexity (e.g. related to the number of parameters) of ZOIP reflects the random Poissonian nature of the disease with other factors, while the lower complexity of the Gamma reflects its more simple nature.

These analyses do not consider dependence on temporal resolution and scale although time and space for stochastic processes relate to each other. The data in this study include yearly case counts; having smaller time units such as months would allow for such consideration Additionally, distributions are assumed to remain constant over the 11 years of observation considering the very minor variations in the inferred distributions that lead to consider VL dynamics at stationary state. Increases have been seen in VL cases over time [54], though case counts between 2000 and 2014 have remained more consistent compared to previous decades [55, 56], indicating that these results are not likely to be sensitive to this assumption. Populations over this time period by municipality generally showed small changes. The mean change in population by municipality was an increase of approximately 11% between 2004 and 2014, and the middle 90% of changes were between a 12% decrease and a 41% increase [37]. These considerations motivate extensions of this study to define the relationship between space and time for scale dependent processes.

Another assumption made in this study is the ability to fit a single probability distribution for VL incidence for the entire nation of Brazil. Since not all of the included states are considered endemic for VL [35], fitting a single distribution for incidence nationwide assumes that the same distribution can represent incidence in both endemic and non-endemic states. However, if conducing a study using VL incidence data, this should be considered in the quantitative analyses that would follow from the results of this study. Other heterogeneities across the nation, such as affluence, urbanization, or climate, which may impact VL incidence, similarly are not considered for distribution fitting but should be accounted for during subsequent analyses.

The results of this study rely on the data collected. VL case data were collected through passive surveillance and notification to the Ministries of Health. It is commonly known that reported cases of infectious diseases only represent a portion of the total cases [57–59], commonly representing the most severe cases. This limits the accuracy of the data, and therefore distribution fitting, by the ability to report cases as well as the potential heterogeneous severity of VL cases. It is also likely that across locations in Brazil, amounts of underreporting of cases differ. The results of this study rely on the assumption that reported cases provide an adequate representation of disease burden. Furthermore, inclusion of both endemic and non-endemic states in the analyses may lead to the inclusion of case data representing both typical VL incidence as well as atypical VL incidence. This could potentially affect distribution fitting if underlying processes leading to typical and atypical incidence differ.

A limitation of this study is the reliance on the criterion for determining differences when comparing distributions and algorithm used for determining best fitting distribution families and parameters. There are numerous methods for performing both tasks, and different methods may lead to slightly different conclusions. The methods of this study do, however, use assumption-free criteria in order to generate the results. A sensitivity analysis was conducted to determine if the number of samples drawn to generate the empirical state-resolution distribution described in section 2.2.1 using Eq (2) might impact RRIG values, and it was found that using 1,000; 2,000; 5,000; and 10,000 samples did not yield distinct difference in RRIG values and no differences in interpretations and conclusions. The threshold choice of 0.05 for the RRIG was an a priori decision. Since this is a continuous value, it used in decision-making in other contexts, other choices for thresholds would be valid.

Another important note is that this study used surveillance units of different sizes, examining aggregation of municipalities of differing land areas and populations and comparisons

among states with different areas and populations. This results from using administrative districts, and still remains useful by using the units recorded in infectious disease surveillance. However, diseases know no political boundaries; yet, an ecosystem-based discretization to define homogeneous high resolution units would be preferable for surveillance such as one based on Digital Elevation Models from which to derive physical ecosystem boundaries that are relevant to disease spread. This would also help the control of diseases to assign to different political entities.

A related topic of research is the existence of spatial autocorrelation in the data. Values of Moran's I using municipality-resolution incidence nationwide showed strong evidence of spatial clustering. Evidence of spatial autocorrelation aligns with the finding that distributional fits for VL incidence are not interchangeable across resolutions and scales. Having cases concentrated in particular local regions would suggest that local factors are important to VL dynamics and should be accounted for in future research. This implies that disease dynamics are local as already highlighted by differing fitted distributions across states, which is consistent with previous works [60, 61]. Any future analyses on VL in Brazil would benefit from the use of methods that account for spatial autocorrelation. For the purposes of distribution fitting, finding distributional families that most accurately characterize incidence is of greater importance than determining a covariance structure that most accurately reflects autocorrelation. Determining clusters and covariance structures is an important component of analysis that follows the results of this study.

## 5. Conclusions

The choice of spatial resolution and scale in infectious disease research is shown to have a potential impact on future results and conclusions when using statistical and mathematical models. The findings from this study should be considered prior to designing quantitative analyses. Finding sensitivity to the spatial resolution and spatial scale of VL surveillance data is of interest to both researchers and government officials for preparedness. Analyses using VL data should consider the findings of this study when planning analyses and controls related to disease processes or population incidence trajectories. Surveillance agencies should note that accurate surveillance by municipality is important because measuring incidence by state alone does not offer an equivalent characterization, and while there do exist small areas with incidences that can describes those of the others, nationwide surveillance at high resolution remains important to consider likely heterogeneity of processes contributing to VL burden. This applies to other diseases with incidences that depend on the scale and resolution of surveillance, which should be examined to assure whether this dependence does exist.

## Supporting information

**S1 Table. Frequency of incidence values by municipality within each state and by state for the nation.**
(CSV)

**S2 Table. Best fitting parameterizations for each considered distribution and associated sum of absolute error (SAE) values.**
(XLSX)

**S1 File. Original data with annual Visceral Leishmaniasis incidence by municipality in Brazil, 2004–2014 with R code used for analysis.**
(ZIP)

## Acknowledgments

The authors thank the Brazilian Ministries of Health for allowing the use of the Visceral Leishmaniasis data for this study. The authors also acknowledge the resources of the Minnesota Supercomputing Institute for computational aid.

## Author Contributions

**Conceptualization:** Joseph L. Servadio, Gustavo Machado, Julio Alvarez, Matteo Convertino.

**Data curation:** Francisco Edilson de Ferreira Lima Júnior, Renato Vieira Alves.

**Formal analysis:** Joseph L. Servadio.

**Funding acquisition:** Julio Alvarez, Matteo Convertino.

**Investigation:** Francisco Edilson de Ferreira Lima Júnior, Renato Vieira Alves.

**Methodology:** Joseph L. Servadio.

**Project administration:** Matteo Convertino.

**Resources:** Gustavo Machado, Julio Alvarez, Francisco Edilson de Ferreira Lima Júnior, Renato Vieira Alves.

**Software:** Joseph L. Servadio.

**Supervision:** Matteo Convertino.

**Validation:** Joseph L. Servadio.

**Visualization:** Joseph L. Servadio.

**Writing – original draft:** Joseph L. Servadio, Gustavo Machado, Julio Alvarez, Matteo Convertino.

**Writing – review & editing:** Joseph L. Servadio, Gustavo Machado, Julio Alvarez, Francisco Edilson de Ferreira Lima Júnior, Renato Vieira Alves, Matteo Convertino.

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
