## [Decision Letter · Decision Letter 0]

31 Mar 2020

PONE-D-20-06883

Information differences across spatial resolutions and scales for disease surveillance and analysis: The case of Visceral Leishmaniasis in Brazil

PLOS ONE

Dear Dr Servadio,

Thank you for submitting your manuscript to PLoS ONE. After careful consideration, we felt that your study has the potential to be published if it is revised to address specific topics raised by two different reviewers. Major concerns were related to MM, for example, criteria for municipalities exclusion, epidemiological data of the Brazilian states; also some supporting information should be included (R code used in the analyses).  For your guidance, a copy of the reviewer' comments was included below. 

We would appreciate receiving your revised manuscript by April 30. To enhance the reproducibility of your results, we recommend that if applicable you deposit your laboratory protocols in protocols.io, where a protocol can be assigned its own identifier (DOI) such that it can be cited independently in the future. For instructions see: http://journals.plos.org/plosone/s/submission-guidelines#loc-laboratory-protocols

We look forward to receiving your revised manuscript.

Kind regards,

Luzia Helena Carvalho, Ph.D.

Academic Editor

PLOS ONE

Journal Requirements:

Reviewers' comments:

Reviewer's Responses to Questions

**Comments to the Author**

1. Is the manuscript technically sound, and do the data support the conclusions?

Reviewer #1: Yes

Reviewer #2: Yes

2. Has the statistical analysis been performed appropriately and rigorously? 

Reviewer #1: Yes

Reviewer #2: Yes

3. Have the authors made all data underlying the findings in their manuscript fully available?

Reviewer #1: Yes

Reviewer #2: Yes

4. Is the manuscript presented in an intelligible fashion and written in standard English?

Reviewer #1: Yes

Reviewer #2: Yes

5. Review Comments to the Author

Reviewer #1: This is a very relevant study that addresses the effect of spatial scales on visceral leishmaniasis surveillance data from Brazil. The manuscript is very well written and understandable. The methods are adequately explained, although some minor details can be added to improve reproducibility.

In infectious disease modelling, fitting disease data to a known statistical distribution is a common first step of the analyses, but oftentimes it is overlooked. The first great contribution of this work to VL modellers is summarised in table 2 – the fitted distributions for country and state-level surveillance datasets. I do not recall seeing the gamma distribution in other VL studies, which usually apply the Poisson distribution. This has the potential to become an important reference for VL modelling exercises in Brazil.

The second main contribution is the effect of spatial scale and the importance of choosing the right dataset for your research question. This has been explored before in theory, but to the best of my knowledge this is the first time I see these scale comparisons in VL data from Brazil.

I recommend the acceptance of the manuscript after some minor details are clarified, specifically:

L125: Explain why the Federal District was not included in the dataset.

L131: Give further info on exclusion criteria for municipalities. Were they excluded due to missing population data only or disease data as well?

L222: It would be very relevant, for reproducibility, if the R code used in the analyses were uploaded as supporting information or uploaded to a code repository such as GitHub.

L289: “different processes driving disease occurrence” – this made me think about the differences between disease occurrence and disease notification, which is the actual data that was analysed. Your discussion might also include comments on how underreporting of VL affect these analyses. It is very likely that underreporting varies among Brazilian states.

Fig. 1: In the map of Brazilian States there are some extra lines appearing where they should not, more specifically inside the states MS, GO, SP, ES and PE. This is probably an artefact of the shapefile that was used, it should be corrected.

Reviewer #2: I thank the authors for this work on an interesting subject. I suggest minor revision according to these comments:

Introduction section:

1. “Urban area” Lines 98 and 101: Please check !

2. Authors don't precise Leishmania species caused VL in Brazil.

MM section:

1. Data collection: Please add subtitle for the first paragraph

2. 22 Brazilian states were included; please add data about epidemiological status: endemic/non endemic state!

Discussion section:

Authors must discuss according to other parameters: epidemiological status of the study area, type of infection....

6. PLOS authors have the option to publish the peer review history of their article (what does this mean?). If published, this will include your full peer review and any attached files.

Reviewer #1: No

Reviewer #2: Yes: Samia BOUSSAA

---

## [Author Response · Author response to Decision Letter 0]

16 Apr 2020

The reviewer comments are seen in quotes, and responses to specific comments are found directly below relevant comments, with each paragraph beginning with the lead author's initials, JLS.

"Reviewer #1: This is a very relevant study that addresses the effect of spatial scales on visceral leishmaniasis surveillance data from Brazil. The manuscript is very well written and understandable. The methods are adequately explained, although some minor details can be added to improve reproducibility.

In infectious disease modelling, fitting disease data to a known statistical distribution is a common first step of the analyses, but oftentimes it is overlooked. The first great contribution of this work to VL modellers is summarised in table 2 – the fitted distributions for country and state-level surveillance datasets. I do not recall seeing the gamma distribution in other VL studies, which usually apply the Poisson distribution. This has the potential to become an important reference for VL modelling exercises in Brazil.

The second main contribution is the effect of spatial scale and the importance of choosing the right dataset for your research question. This has been explored before in theory, but to the best of my knowledge this is the first time I see these scale comparisons in VL data from Brazil.

I recommend the acceptance of the manuscript after some minor details are clarified, specifically:

L125: Explain why the Federal District was not included in the dataset."

JLS: This study largely aimed to compare distributional fits at spatial resolutions, using incidence by municipality and by state. Because the Federal District is not a state with multiple municipalities, Visceral Leishmaniasis incidence cannot be aggregated to allow municipality-resolution and state-resolution incidence to be calculated. This clarification has been added to the text on line 133 of the revised manuscript.

"L131: Give further info on exclusion criteria for municipalities. Were they excluded due to missing population data only or disease data as well?"

JLS: Excluded municipalities were not present in the IBGE population data. This sentence was rewritten to make this clearer, and appears on line 140 of the revised manuscript.

"L222: It would be very relevant, for reproducibility, if the R code used in the analyses were uploaded as supporting information or uploaded to a code repository such as GitHub."

JLS: The third supplemental material, originally containing the original data, has been replaced with S3_File.zip. This contains a folder with the original incidence data as well as the R code used for these analyses.

"L289: “different processes driving disease occurrence” – this made me think about the differences between disease occurrence and disease notification, which is the actual data that was analysed. Your discussion might also include comments on how underreporting of VL affect these analyses. It is very likely that underreporting varies among Brazilian states."

JLS: This is a very important note. This fact is very present in infectious disease literature. A paragraph has been added to the discussion highlighting that this study uses reported cases, which represent a subset of the true number of cases. This paragraph begins on line 418 of the revised document.

"Fig. 1: In the map of Brazilian States there are some extra lines appearing where they should not, more specifically inside the states MS, GO, SP, ES and PE. This is probably an artefact of the shapefile that was used, it should be corrected."

JLS: We appreciate catching this detail. This has been fixed.

"Reviewer #2: I thank the authors for this work on an interesting subject. I suggest minor revision according to these comments:

Introduction section:

1. “Urban area” Lines 98 and 101: Please check !"

JLS: The citations associated with the statements indicate that Visceral Leishmaniasis has a canine reservoir in Brazil and that the disease has been seen in urban settings and is associated with urbanization in recent decades, in contrast to having been seen in rural settings previously. The wording has changed to reflect that canine reservoirs are not exclusively in urban settings and that the recent shift to greater incidence in urban settings occurred since 1980 rather than the most recent years. 

"2. Authors don't precise Leishmania species caused VL in Brazil."

JLS: Cases of VL in Brazil have been attributed to L. infantum. This has been added on line 90 of the revised manuscript.

"MM section:

1. Data collection: Please add subtitle for the first paragraph"

JLS: This has been added.

"2. 22 Brazilian states were included; please add data about epidemiological status: endemic/non endemic state!"

JLS: Classifications of states as endemic/non endemic according to the Brazilian Ministries of Health, with citations, have been added, starting on line 104 of the revised manuscript as well as in Table 2.

"Discussion section:

Authors must discuss according to other parameters: epidemiological status of the study area, type of infection...."

JLS: This point of epidemiological status has been included. Similarly, other heterogeneities across Brazil are not accounted for in the distribution fitting process, but should be considered in any modeling studies that follow. This has been added in the paragraph starting on line 409 of the revised manuscript as well as on lines 424-428.

JLS: The point of the type of infection connects with the other reviewer’s comment regarding underreporting of cases, as the most severe cases are less commonly underreported. The severity of infection has been added as a point describing underreporting on lines 446-448 of the revised manuscript.

---

## [Decision Letter · Decision Letter 1]

29 May 2020

PONE-D-20-06883R1

Information differences across spatial resolutions and scales for disease surveillance and analysis: The case of Visceral Leishmaniasis in Brazil

PLOS ONE

Dear Dr. Servadio,

Thank you for submitting your manuscript for review to PLoS ONE. After careful consideration, we feel that your manuscript will likely be suitable for publication if the authors revise it to address few points raised now by the reviewers. According to reviewers, there are some specific areas where further improvements would be of substantial benefit to the readers, including data analysis. 

We look forward to receiving your revised manuscript.

Kind regards,

Luzia Helena Carvalho, Ph.D.

Academic Editor

PLOS ONE

Reviewers' comments:

Reviewer's Responses to Questions

**Comments to the Author**

1. If the authors have adequately addressed your comments raised in a previous round of review and you feel that this manuscript is now acceptable for publication, you may indicate that here to bypass the “Comments to the Author” section, enter your conflict of interest statement in the “Confidential to Editor” section, and submit your "Accept" recommendation.

Reviewer #2: All comments have been addressed

Reviewer #3: All comments have been addressed

Reviewer #4: All comments have been addressed

2. Is the manuscript technically sound, and do the data support the conclusions?

Reviewer #2: Yes

Reviewer #3: Yes

Reviewer #4: Yes

3. Has the statistical analysis been performed appropriately and rigorously? 

Reviewer #2: Yes

Reviewer #3: Yes

Reviewer #4: Yes

4. Have the authors made all data underlying the findings in their manuscript fully available?

Reviewer #2: Yes

Reviewer #3: Yes

Reviewer #4: Yes

5. Is the manuscript presented in an intelligible fashion and written in standard English?

Reviewer #2: Yes

Reviewer #3: Yes

Reviewer #4: Yes

6. Review Comments to the Author

Reviewer #2: The authors have adequately addressed my comments. This version of the manuscript is now acceptable for publication.

Reviewer #3: This study is a good study and it is recommended to be accepted for publish. some comments:

It is better to work on the resolution of the maps.

Specify the north and south directions on the map.

In the introduction, the mortality rate of visceral leishmaniasis in the world and Brazil should be mentioned.

Reviewer #4: I would like to thank the authors for this interesting work on VL. Please find my detailed comments below:

The Modifiable Areal Unit Problem (MAUP) is a common problem for analyzing spatial data, which I think the authors have partially engaged by mentioning about the scale invariance problem. However, it is still unclear from the introduction as to where the research gap lies, when it is a common knowledge that changing the scale is highly likely to change the results or findings of a spatial analysis. The objective of the work was outlined well but the motivation of the work requires more clarity.

Line 145 to 146: Any reason why this decision was taken?

Line 152: What exploratory analyses were conducted and what criteria for suitability were set for choosing the distributions in Table 1? I see ZIP and ZOIP are used, which is understandable given the count nature of the data. But it is unclear as to why the exponential, gamma etc. models were necessary. This part of the manuscript could be a little tricky to deal with, as the justifications of the selection of the distributions need to be provided without making the methods too technical for the readers to understand. Again, the justification needs to be clear enough so that any readers with a geostatistical background can understand the rationale behind the selection and can benefit from this work.

Line 165: How was “()” or “the probability of observing an incidence o f cases per 100,000 person-years” determined? Did the process involve the computation of any expected number of cases? If yes, what method was applied for the computation?

Line 187: The authors have repeatedly mentioned this (or similar) term “inferential conclusions”. I think it is better to clearly outline what this term entails for the readers to understand the methods better.

The left hand side of Eq2 is absent

Line 223: Any justification for this 5% threshold? Please include in the text for strengthening your selection.

Results have been presented well but the discussion needs to engage with the issue of spatial heterogeneity that has not been incorporated into the analysis. For example, Fig 1 shows the distribution of VL cases can possibly exhibit spatial clustering and even spatial autocorrelation. Under these circumstances, there is a good possibility that the incidence rates might not match across different scales, as the disease burden is not distributed equally across all the municipalities.

7. PLOS authors have the option to publish the peer review history of their article (what does this mean?). If published, this will include your full peer review and any attached files.

Reviewer #2: Yes: Samia BOUSSAA

Reviewer #3: Yes: Eslam Moradi-Asl

Reviewer #4: No

---

## [Author Response · Author response to Decision Letter 1]

5 Jun 2020

Review comments are below. Responses to comments are found directly under the comments, and all paragraphs with responses are indented and designated with the corresponding author's initials (JLS). 

Reviewer #2: The authors have adequately addressed my comments. This version of the manuscript is now acceptable for publication.

 JLS: We appreciate the reviewer’s attention to our revision.

Reviewer #3: This study is a good study and it is recommended to be accepted for publish. some comments:

It is better to work on the resolution of the maps.

 JLS: All figures have an improved resolution.

Specify the north and south directions on the map.

 JLS: The maps in Figures 2 and 4 contain a compass arrow pointing to North in the lower left corner of each map. Figure 1 does not contain this because it is a conceptual diagram; specific details of the orientation of Brazil are not crucial to its interpretation. 

In the introduction, the mortality rate of visceral leishmaniasis in the world and Brazil should be mentioned.

 JLS: This has been added. The global mortality rate, represented via estimates of total estimated cases and deaths, is reported on lines 102 and 103 of the revised manuscript. The estimated mortality rate in Brazil is reported on lines 107-110.

Reviewer #4: I would like to thank the authors for this interesting work on VL. Please find my detailed comments below:

The Modifiable Areal Unit Problem (MAUP) is a common problem for analyzing spatial data, which I think the authors have partially engaged by mentioning about the scale invariance problem. However, it is still unclear from the introduction as to where the research gap lies, when it is a common knowledge that changing the scale is highly likely to change the results or findings of a spatial analysis. The objective of the work was outlined well but the motivation of the work requires more clarity.

 JLS: This study’s relevance to the MAUP is related to that to ecological fallacy. Few studies have quantified this, as this concept is intuitive. This study offers an example of a real quantification of this issue. These points are included in the introduction on lines 63-66 and in the discussion on lines 366-368 of the revised manuscript. 

 JLS: The second motivation and objective of this study was to examine scale invariance, which differs in that the size of the entire area considered is examined, while keeping the areal units constant. This is motivated by the ability to use a subset of the nation of Brazil to describe the nation’s VL burden, as described in the introduction. 

Line 145 to 146: Any reason why this decision was taken?

 JLS: Previous studies exploring this have done so by first posing a substantive research question and then following up by repeating analyses. The argument made in this manuscript is that, while providing some insight, the repeated analysis is subject to the assumptions of the model. Because this study aimed to address the question of scale/resolution sensitivity as the primary objective, a model using other covariates was not wanted. Fitting the distributional fit represents the simplest way of describing disease incidence and would directly relate to either a stochastic node of a mathematical model or a link function of a statistical model. An additional sentence emphasized this point on lines 163-166 of the revised manuscript.

Line 152: What exploratory analyses were conducted and what criteria for suitability were set for choosing the distributions in Table 1? I see ZIP and ZOIP are used, which is understandable given the count nature of the data. But it is unclear as to why the exponential, gamma etc. models were necessary. This part of the manuscript could be a little tricky to deal with, as the justifications of the selection of the distributions need to be provided without making the methods too technical for the readers to understand. Again, the justification needs to be clear enough so that any readers with a geostatistical background can understand the rationale behind the selection and can benefit from this work.

 JLS: The exploratory analyses were primarily visualizations and examining summary statistics such as quantiles and moments. Suitability refers to a distribution having a nonnegative support. The purpose of selecting a variety of distributions was to consider many common distributional families to explore the possibility of one providing the closest fit to the data. These points were made more explicit in the text and appear on lines 172-173 of the revised manuscript. 

Line 165: How was “()” or “the probability of observing an incidence o f cases per 100,000 person-years” determined? Did the process involve the computation of any expected number of cases? If yes, what method was applied for the computation?

 JLS: The values of p(x) were calculated using the formulas in Table 1. These were evaluated through base R functions such as rpois(), runif(), etc, based on the distributional family being evaluated. In the cases of the Power Law distribution, the ‘poweRlaw’ package was used as cited at the end of the methods section.

Line 187: The authors have repeatedly mentioned this (or similar) term “inferential conclusions”. I think it is better to clearly outline what this term entails for the readers to understand the methods better.

 JLS: We appreciate this point. The intended message is that the finding that the scale and resolution choices can change the distributional fit of incidence implies that any statistical or mathematical models using this data could suffer from similar sensitivity given choices of scale and resolution. The wording has been changed throughout the manuscript to explicitly state this every time with consistent wording. Notable changes include those on lines 210-213 and 437-438 of the revised manuscript.

The left hand side of Eq2 is absent

 JLS: A value has been added for the left hand side. It is also referenced in a sentence added to the end of the paragraph (line 232) to clarify the methods used.

Line 223: Any justification for this 5% threshold? Please include in the text for strengthening your selection.

 JLS: The 5% threshold was decided a priori, and not based on previous use. This has been added on line 251 of the revised manuscript and a sentence has been added to the discussion on lines 452-464 that states this explicitly.

Results have been presented well but the discussion needs to engage with the issue of spatial heterogeneity that has not been incorporated into the analysis. For example, Fig 1 shows the distribution of VL cases can possibly exhibit spatial clustering and even spatial autocorrelation. Under these circumstances, there is a good possibility that the incidence rates might not match across different scales, as the disease burden is not distributed equally across all the municipalities.

 JLS: The discussion of spatial heterogeneity has been expanded to incorporate these points more directly. This was noted in the last paragraph of the discussion section. It has been made into its own paragraph with an addition to make this point more explicit.

 JLS: In addition to these specific comments, the manuscript has been checked for any spelling/grammatical errors as well as citations and other formatting matters.

---

## [Decision Letter · Decision Letter 2]

25 Jun 2020

Information differences across spatial resolutions and scales for disease surveillance and analysis: The case of Visceral Leishmaniasis in Brazil

PONE-D-20-06883R2

Dear Dr. Servadio,

We’re pleased to inform you that your manuscript has been judged scientifically suitable for publication and will be formally accepted for publication once it meets all outstanding technical requirements.

Kind regards,

Luzia Helena Carvalho, Ph.D.

Academic Editor

PLOS ONE

Additional Editor Comments (optional):

Reviewers' comments:

Reviewer's Responses to Questions

**Comments to the Author**

1. If the authors have adequately addressed your comments raised in a previous round of review and you feel that this manuscript is now acceptable for publication, you may indicate that here to bypass the “Comments to the Author” section, enter your conflict of interest statement in the “Confidential to Editor” section, and submit your "Accept" recommendation.

Reviewer #2: All comments have been addressed

Reviewer #3: All comments have been addressed

Reviewer #4: All comments have been addressed

2. Is the manuscript technically sound, and do the data support the conclusions?

Reviewer #2: Yes

Reviewer #3: Yes

Reviewer #4: Yes

3. Has the statistical analysis been performed appropriately and rigorously? 

Reviewer #2: Yes

Reviewer #3: Yes

Reviewer #4: Yes

4. Have the authors made all data underlying the findings in their manuscript fully available?

Reviewer #2: (No Response)

Reviewer #3: Yes

Reviewer #4: Yes

5. Is the manuscript presented in an intelligible fashion and written in standard English?

Reviewer #2: (No Response)

Reviewer #3: Yes

Reviewer #4: Yes

6. Review Comments to the Author

Reviewer #2: (No Response)

Reviewer #3: My dear authors ,

All comments have been answered and addressed.

Thanks for the great work and acceptable.

Best regards

Reviewer #4: (No Response)

7. PLOS authors have the option to publish the peer review history of their article (what does this mean?). If published, this will include your full peer review and any attached files.

Reviewer #2: **Yes: **Samia BOUSSAA

Reviewer #3: **Yes: **Dr.Eslam Moradi-Asl

Reviewer #4: No

---

## [Editor Report · Acceptance letter]

6 Jul 2020

PONE-D-20-06883R2 

Information differences across spatial resolutions and scales for disease surveillance and analysis: The case of Visceral Leishmaniasis in Brazil 

Dear Dr. Servadio:

I'm pleased to inform you that your manuscript has been deemed suitable for publication in PLOS ONE. Congratulations! Your manuscript is now with our production department. 

Kind regards, 

on behalf of

Dr. Luzia Helena Carvalho 

Academic Editor

PLOS ONE